# Synergistic Driver-Laser/Modulator Co-Design with Versatile Output Stage: A Unified Optical Transmitter EIC Design Approach

**DOI:** 10.3390/mi16111262

**Published:** 2025-11-06

**Authors:** Ruixuan Yang, Cailing Li, Yifei Xia, Yuye Yang, Li Geng, Dan Li

**Affiliations:** Faculty of Electronic and Information Engineering, Xi’an Jiaotong University, Xi’an 710049, China; yrxay358@stu.xjtu.edu.cn (R.Y.); 2204313340@stu.xjtu.edu.cn (C.L.); xyf1596320@stu.xjtu.edu.cn (Y.X.); 4121156016@stu.xjtu.edu.cn (Y.Y.); gengli@xjtu.edu.cn (L.G.)

**Keywords:** low power, optical transmitter, CMOS, driver, reconfigurable back termination

## Abstract

With the rapid deployment of artificial intelligence (AI) data centers, demand for optical modules surges—alongside faster upgrades and stricter low-power requirements. However, traditional optical driver integrated circuits (ICs) rely on device-specific customization, which lengthens driver design cycles, delays module deployment, and raises costs, becoming a bottleneck for optical module evolution. To address these issues, this work proposes a unified optical transmitter electronic integrated circuit (EIC) design approach featuring synergistic driver-laser/modulator co-design and a versatile output driver (VOD). The VOD can be configured into three output impedance states (open-drain, differential 50-Ω, or differential 100-Ω), enabling it to drive various optical devices like distributed feedback lasers (DFBs), vertical-cavity surface-emitting lasers (VCSELs), electro-absorption modulated lasers (EMLs), and Mach-Zehnder modulators (MZMs) with a single design, minimizing device-specific customization. Meanwhile, its power consumption is also adjustable to maximize the power efficiency. The proposed design approach demonstrates the potential to address the critical interoperability, cost, and power challenges faced by AI data centers, providing a scalable template for next-generation coherent and 4-level pulse amplitude modulation systems and facilitating rapid deployment.

## 1. Introduction

Optical communication systems have become the backbone of modern high-speed communication networks, owing to their exceptional advantages including ultra-high bandwidth capacity, extremely low transmission loss, and superior immunity to electromagnetic interference. These systems are extensively deployed in various applications ranging from long-haul fiber-optic backbones to data center interconnects and 5G fronthaul/backhaul networks [1,2,3,4,5].

In particular, the rapid expansion of artificial intelligence (AI) data centers—driven by exponentially growing computational demands—imposes two critical challenges on optical interconnect technologies: increasingly accelerated deployment cycles for optical modules and a stringent need for lower power consumption. These systems must not only support higher data rates and greater port densities but also comply with strict thermal and energy constraints. Effectively addressing both performance and efficiency is therefore essential to sustain the development of next-generation AI infrastructure.

At the heart of these systems, optical transmitters serve as critical components that convert electrical signals into optical signals. Within the transmitter architecture, the driver circuit plays a pivotal role in ensuring optimal performance. Its primary functions include:(1)Signal amplification: The driver significantly boosts the amplitude of the input electrical signal to meet the driving voltage/current swing requirements of the subsequent optical modulator.(2)Impedance matching: It carefully maintains proper impedance matching throughout the signal path to minimize reflections and inter-symbol interference (ISI) and ensure maximum power transfer.(3)Signal conditioning: The driver often incorporates additional features such as signal shaping and pre-emphasis to compensate for bandwidth limitations in the transmission path.
The design of the driver circuit must be meticulously tailored to the specific characteristics of the optical modulator it interfaces with, including the modulator type, the packaging forms, the required driving voltage/current swing and the bandwidth specifications.

Previous research has primarily focused on optimizing optical transmitters within specific domains—such as power efficiency enhancement [6], linearity improvement [7], and bandwidth expansion [8]—with each typically tailored to a single application scenario. However, these designs often suffer from strong dependencies on specific optical devices, which result in extended design cycles, slow iteration, and limited scalability. Their inability to adapt to varying voltage, current, and impedance requirements across different modulators further leads to suboptimal power efficiency and increased operational power consumption. Additionally, the lack of reconfigurability impedes hardware reuse, elevating both development costs and system complexity. Therefore, a versatile transmitter architecture is highly desirable in modern optical communication systems.

To address the impedance matching issues of different optical devices, this paper presents a versatile output driver (VOD) integrated circuit (IC) design approach that respectively adopts passive (PBT) and active back termination (ABT) techniques. The proposed architecture supports three manually selectable impedance modes—open-drain, differential 50-Ω, and differential 100-Ω back termination—enabling optimized driving of both directly and externally modulated lasers with diverse packaging requirements. Two 10-Gbps driver prototypes implementing the PBT and ABT schemes have been fabricated in a standard 180-nm complementary metal-oxide-semiconductor (CMOS) process, demonstrating cost-effective manufacturability while ensuring compatibility with various optical transceivers.

## 2. Conventional Driver Design

As mentioned above, conventional driver designs must be meticulously tailored to the specific characteristics of the optical modulator it interfaces with. This section will briefly introduce the customized design requirements for drivers from the aspects of the modulator types and the packaging forms.

Modulators are generally categorized into two types: direct modulation and indirect (or external) modulation. Directly-modulated lasers DMLs, such as distributed feedback lasers (DFBs) and vertical-cavity surface-emitting lasers (VCSELs), operate by modulating the current supplied to the laser diode, causing the output light intensity to vary with the electrical signal. They are widely used in short-reach optical communication systems, like local area networks (LANs) and data centers, where simplicity, low cost, and compact design are prioritized [9]. In contrast, externally-modulated lasers, including electroabsorption modulated lasers (EMLs) and Mach-Zehnder modulators (MZMs), employ a continuous-wave (CW) laser operating at a constant power, while an external modulator driven by voltage is used to encode the data onto the laser. Externally-modulated lasers are preferred for long-distance communication in metropolitan or long-haul networks, where the signal integrity and stability are essential [10]. Table 1 summarizes the similarities and differences in the typical characteristics of these modulators [11]. In simple terms, a directly-modulated laser is generally driven by a current-mode driver, while an externally-modulated laser is generally driven by a voltage-mode driver. Furthermore, the required driving swing amplitude varies across different devices, imposing distinct requirements on the output stage of the driver.

Packaging is another consideration. Vendors often package lasers and modulators within a Transmitter Optical Sub-Assembly (TOSA) due to its compact design and robustness. To facilitate ease of use and interchangeability, users and suppliers have established a Multi-Source Agreement (MSA) defining the dimensions, pin configurations, and impedance control demands for TOSA [12]. As shown in Figure 1a, the input impedance of an EML-based TOSA is defined as a single-ended 50 Ω. In contrast, for a DML-based TOSA, the driver output impedance is specified as a single-ended 25 Ω or differential 50 Ω, while the TOSA input impedance and transmission line characteristic impedance are left undefined, as shown in Figure 1b. This flexibility enables vendors to optimize performance across different DML implementations. For specific optical device packaging, transistor outline (TO) packages are widely adopted, offering advantages such as low cost, hermetic sealing, efficient thermal dissipation, and precise optical alignment capabilities.

Besides the TOSA package, co-packaged optics (CPO) is another approach for packaging lasers and modulators, as shown in Figure 2, and is regarded a development trend for data center optical interconnects due to its higher bandwidth and lower power consumption. By integrating photonic integrated circuits (PICs) and electronic integrated circuits (EICs) on a shared substrate, the CPO solution enables shorter electrical interconnection lengths, allowing for impedance mismatch between the driver and modulator [13].

## 3. Versatile Output Driver Design

Based on the distinct output impedance requirements of conventional drivers for different optical devices—such as 25 Ω for DMLs, 50 Ω for EMLs, and open-drain (OD) driving for some MZMs—we have developed a VOD design, illustrated in Figure 3, capable of supporting multiple types of optical devices.

The versatile output driver consists of an input matching network, a pre-driver and a versatile output stage. The pre-driver employs a two-stage amplifier configuration with uniform gain. To adequately drive the substantial capacitive load of the output stage, the second stage is designed with twice the transistor size and current consumption of the first. This scaling halves the output impedance of the second stage, thereby pushing the pole associated with its output resistor and the input capacitor of the output stage to a higher frequency. Furthermore, the bandwidth is extended toward the process limit of the 180 nm node by incorporating a shunt-peaking inductor. The versatile output stage supports switching among multiple output impedance modes, including open drain (OD), differential 50 Ω, and differential 100 Ω, while also allowing tail current adjustment to achieve the required output swing and maintain low power consumption. It can be implemented using two approaches: passive back termination (PBT) and active back termination (ABT). The designs utilizing these two approaches will be presented in the following sections.

### 3.1. PBT Based VOD

As illustrated in Figure 4, passive back termination can be achieved using various components, including a diffused/polysilicon/metal resistor, a metal-oxide-semiconductor field-effect transistor (MOSFET) operating in the linear region, or a diode-connected MOSFET.

The resistor’s linear model is particularly convenient because its resistance is easily calculated from the foundry’s sheet resistance parameter and the designer-specified length and width dimensions. When making a specific selection, the primary considerations are its temperature dependence, relative accuracy, and the area required to meet current density limits. Table 2 compares the essential characteristics of the three resistor types.

Generally, silicided diffused resistors, with a typical sheet resistance of approximately 20 Ω per square, are suitable for implementing low-value resistors, offering high current density tolerance and inherent isolation via reverse-biased p-n junctions, enabling multiple resistors within a single tub for area efficiency.

Unsilicided polysilicon resistors provide higher sheet resistance, lower temperature drift, and better accuracy [14], making them ideal for high-value resistors, though their limited current tolerance restricts high-current use.

Metal resistors excel in precision and stability, with laser trimming further enhancing accuracy. Nevertheless, their inherently low sheet resistance makes them unsuitable for high-value resistor implementation. Even when realizing a modest 50-Ω resistor, the required area becomes prohibitively large to maintain adequate current density tolerance.

A MOSFET operating in the linear region can also serve as a passive back termination load, yet its low resistance is constrained by operating conditions. Its limited drain-to-source voltage ($V_{\mathrm{DS}}$) restricts the achievable output swing while maintaining proper back termination, while pronounced nonlinearity makes it unsuitable for high-linearity applications.

The diode-connected MOSFET provides an alternative, with termination defined by its transconductance ($g_{m}$), offering improved linearity by tracking the gain transistor’s $g_{m}$. However, it introduces an inevitable voltage drop of gate-to-source voltage $V_{\mathrm{GS}}$ and suffers from breakdown voltage limitations, constraining output swing. Moreover, its impedance varies with drive current, potentially impairing matching under large-signal operations.

Figure 5 shows a 6-dB-gain amplifier with various load configurations and their DC operating points. The simulated 3-dB bandwidths are 6.1 GHz, 5.5 GHz, and 5.4 GHz, respectively, reflecting the increased parasitic capacitance from MOSFET-based loads. The simulated total harmonic distortion (THD) versus output swing is also provided in Figure 5d. The load employing linear-region MOSFETs achieves the lowest power consumption but exhibits the highest THD. In contrast, while the diffused resistor and diode-connected MOSFET loads yield comparable THD performance, the latter consumes more power with higher power supply requirement shown in Figure 5c, confirming prior analysis. Despite process and temperature sensitivity, diffused resistors maintain the best linearity–efficiency trade-off among passive termination options.

The first design implements passive back termination, as illustrated in Figure 6. To minimize power dissipation, an external bias-T network is used to supply DC bias to the output stage, thereby avoiding additional on-chip power consumption. The reconfigurable output impedance is achieved through a network of P-channel MOSFET (PMOS) switches that selectively connect or disconnect termination resistors. By controlling the ON/OFF state of these switches, the effective termination resistance can be digitally adjusted to support multiple operating modes.

In MOSFET-based differential pair amplifiers operating in saturation region, the square-law behavior yields the following analytical expressions:(1)$$\left\{\begin{array}{cc} Gain & =2g_{m}R_{L}\\ \frac{1}{2}I_{\mathrm{ss}} & =\frac{1}{2}K\frac{W}{L}V_{\mathrm{OD}}^{2}\\ g_{m} & =K\frac{W}{L}V_{\mathrm{OD}}\\ Swing & =2I_{\mathrm{ss}}R_{L} \end{array}\right.$$(2)$$\Rightarrow \left\{\begin{array}{cc} Gain & =2\sqrt{K\frac{W}{L}I_{\mathrm{ss}}}R_{L}\\ Swing & =2I_{\mathrm{ss}}R_{L} \end{array}\right.$$
where *K* is is a coefficient determined by the type of MOSFET and the process node, *W* and *L* are respectively the channel width and length of the MOSFET, $I_{\mathrm{ss}}$ is the tail current, and $R_{L}$ is the load of differential pair.

From Equation (Equation 2), modifying the terminal resistances without compensatory circuit adjustments inherently affects both the output swing amplitude and voltage gain characteristics. To address this, we can maintain a consistent swing by modifying the tail current $I_{\mathrm{SS}}$, as demonstrated in Figure 6b. By scaling $I_{\mathrm{SS}}$ proportionally with $R_{L}$ variations, we maintain constant swing amplitude. Nevertheless, adjusting the tail current $I_{\mathrm{SS}}$ alone does not keep the consistent gain, as shown in Figure 7a. If the same gain is required to avoid imposing higher swing requirements on the signal source, additional gain must be introduced at the pre-driver stage. This approach, however, not only increases system complexity and undermines low-power operation, but also further constrains the maximum achievable bandwidth of the entire driver.

To ensure both consistent swing and gain, the NMOS width (W) must be simultaneously adjusted to preserve a constant current density. Therefore, the circuit topology shown in Figure 6a is preferred, where the input differential pair is divided into two sections, and the tail current is distributed according to the current density at the peak transit frequency of the NMOS. The three operating modes are illustrated in Figure 8. This approach effectively maintains consistent gain and swing, as shown in Figure 7b, while preserving the maximum achievable bandwidth of the system, ensuring optimal performance under varying conditions.

The performance of the driver with PBT is demonstrated in Figure 9. The open-drain mode presents the lowest bandwidth of 4.6 GHz, which is caused by its highest total load impedance, while delivering the largest differential peak-to-peak output swing of 3 V_ppd_ with a input swing of 800 mV_ppd_ and a power consumption of 180 mW. The differential 50-Ω mode exhibits approximately 3 dB higher gain than the differential 100-Ω mode. This difference arises because the vector network analyzer (VNA) presents a differential 100-Ω load for the driver, rather than a differential 50-Ω termination. In terms of output swing, the 50-Ω mode reaches 2.0 V_ppd_, compared to 1.5 V_ppd_ for the 100-Ω mode. Additionally, the 50-Ω mode consumes 240 mW of power, whereas the 100-Ω mode consumes 180 mW. The originally measured differential to differential S22 parameters under the VNA’s differential 100-Ω loading condition are shown in Figure 9c for both the differential 50-Ω and 100-Ω modes, with the results consistent with their respective output impedance characteristics. The 100-Ω mode demonstrates good impedance matching, achieving a return loss better than 10 dB across a 7-GHz bandwidth. The differential-to-differential output return loss scattering parameter S22 of the 50-Ω mode under a 50-Ω differential load can be renormalized using Equation (Equation 3), as illustrated in Figure 10, which also demonstrates good impedance matching.(3)$$\left\{\begin{array}{cc} Z_{dd22} & =100\frac{1+S_{dd22}^{100\Omega }}{1-S_{dd22}^{100\Omega }}\\ S_{dd22}^{50\Omega } & =\frac{Z_{dd22}-50}{Z_{dd22}+50} \end{array}\right.$$
where $S_{dd22}^{100\Omega }$ is the measured differential-to-differential output return loss scattering parameter, $Z_{dd22}$ is the calculated differential-to-differential output impedance, and $S_{dd22}^{50\Omega }$ is the resulted output return loss scattering parameter under a 50-Ω differential load.

### 3.2. ABT Based VOD

Passive back termination offers a simple and broadband termination solution. However, it consumes twice the power to achieve the same swing. As reported in [6], active back termination provides effective back-termination without reducing the output swing, potentially saving up to 50% of power.

The fundamental implementation of the driver output stage with ABT circuit is illustrated in Figure 11, which consists of:(1)An open-drain main driver (MDRV) with tail current $I_{\mathrm{mod}}$;(2)An ABT circuit comprising: A reduced tail current amplifier ($I_{\mathrm{mod}}/K$); Raised resistive loads ($KR_{D}$) to maintain equivalent voltage swing; And a low-output-impedance buffer (BUF) with termination resistor $R_{T}$.
The design ensures no forward output current flows through $R_{T}$ during operation. Only reflected signals are absorbed by the termination resistor $R_{T}$. While increasing the proportional coefficient *K* improves driving efficiency, this enhancement directly trades off against bandwidth reduction, necessitating careful optimization between these competing parameters. Under nominal operating conditions, the proportional coefficient *K* is typically set to approximately 10 to optimize power efficiency while maintaining acceptable bandwidth. This method is well-suited for future high-density interconnect networks, where power consumption is a critical concern.

The schematic of the proposed reconfigurable output stage with ABT is shown in Figure 12, consisting of a MDRV and an Cherry-Hopper-based active back termination cicuitry.

Ignoring the common mode feedback resistors ($R_{\mathrm{CMFB}}$), the drain-to-source resistance ($r_{\mathrm{ds},MP1,2}$) and the output resistance of casode transistors MN1-MN4 which are sufficiently large, the differential output impedance $Z_{dd22}$ can be approximated as(4)$$Z_{dd22}\approx \frac{2}{g_{m,MN5,6}}$$
where the $g_{m,MN5,6}$ is the transconductance of transitors MN5 and MN6, which corresponds to the size of transitors MN5 and MN6 and their tail current. Thus, the output impedance can be adjusted by tuning the tail current $I_{SS2}$.

Additionally, the tail current $I_{SS1}$ and $I_{SS3}$ need to be set proportionally based on the desired output impedance $Z_{0}$ and the feedback resistors $R_{\mathrm{FB}}$, as shown in the following Equation (Equation 5).This configuration maintains constant gate voltages for transistors MN5 and MN6, enabling high driving efficiency. The driving efficiency η can be calculated using Equation (Equation 6).(5)$$K=\frac{I_{SS3}}{I_{SS1}}=1+\frac{R_{\mathrm{FB}}}{Z_{0}}$$
where *K* is the proportion of $I_{SS3}$ and $I_{SS1}$
(6)$$\eta =\frac{I_{SS3}-I_{SS1}}{I_{SS3}}=\frac{R_{\mathrm{FB}}}{R_{\mathrm{FB}}+Z_{0}}$$
A larger $R_{\mathrm{FB}}$ improves driving efficiency. However, it comes with a trade-off in bandwidth. In this work, the proportion K is set to 12, achieving a driving efficiency of up to 91.6%. And the tail current $I_{SS2}$ is set to approximately 25% of the maximum tail current $I_{SS3}$, contributing to a maximum total efficiency of 77.4% for the output stage.

Figure 13 shows the simulated result of the proposed ABT based versatile output driver. This confirms the previous analysis of ABT, which changes the output impedance without affecting the output swing.

The driver performance incorporating ABT is shown in Figure 14. Compared to the PBT based VOD, the ABT based design exhibits lower bandwidth, which can be attributed to its more complex circuitry and associated parasitic capacitance. Nevertheless, the independence between output swing and output impedance is clearly validated by eye diagram measurements, with the ABT-based VOD achieving 3.4 V_ppd_ in differential 50-Ω mode, 3.6 V_ppd_ in differential 100-Ω mode, and 4.0 V_ppd_ in open-drain mode. Impedance matching performance is confirmed through S22 measurements. The differential-to-differential S22 parameter of the 50-Ω mode under a 50-Ω differential load is also renormalized using Equation (Equation 3), as shown in Figure 15, demonstrating good impedance matching performance.

Although the ABT-based VOD exhibits a narrower matching bandwidth due to increased parasitic effects, its superior power efficiency makes it particularly suitable for high-density, power-sensitive systems implemented in advanced processes with adequate transit frequency.

## 4. Discussions

The previous section has introduced the design of two versatile output drivers (VODs) based on PBT and ABT architectures, demonstrating a unified optical transmitter EIC approach that supports three impedance modes and enables interoperability across various optical transmitter types without requiring device-specific customization. A comparative analysis of these two VOD implementations and their application scenatios will be presented in the following section.

As shown in Table 3, the PBT-based versatile output driver offers programmable output swing, simple design, and wide bandwidth, demonstrated by 10-Gbps clear eye diagrams across all impedance modes. Its reconfigurable termination resistor network and fixed current density design of the output stage maintain consistent gain and swing. However, it suffers from a fundamental 50% power efficiency limit due to power dissipation in the termination resistor. Thus, it is best suited for high-speed or long-reach interconnects where bandwidth is prioritized over power efficiency.

In contrast, the ABT-based driver achieves significantly higher power efficiency, with a total output stage efficiency of up to 77.4%, by employing a Cherry-Hooper active termination that only dissipates power for reflected signals. It also maintains a nearly constant output swing across modes. The primary trade-offs are increased circuit complexity and reduced bandwidth. Consequently, ABT is ideal for power-sensitive and high-density applications where sufficient bandwidth is available.

The choice between PBT and ABT thus involves a direct trade-off among power efficiency, bandwidth, and design complexity. For systems targeting the highest possible data rates with simpler implementation, PBT remains a robust and reliable choice. For systems where power consumption is the dominant constraint, such as in next-generation AI cluster interconnects and CPO-based switches, ABT offers a compelling advantage, albeit with more stringent design and stability considerations.

Furthermore, the reconfigurable core presented in this work—supporting open-drain (OD), differential 50-Ω, or differential 100-Ω modes—proves adaptable to both termination styles. This demonstrates the viability of a unified driver platform that can be tailored towards either efficiency or bandwidth supremacy based on the termination methodology. The open-drain mode, usable with both PBT and ABT, is particularly future-proof for CPO and short-reach scenarios where impedance mismatch can be tolerated, enabling further power savings.

Table 4 summarizes the performances comparison with the state of the art. To quantify the output efficiency, we introduce the following figure of merit (FoM): (7)$$\mathrm{FoM}=\frac{I_{\mathrm{mod},\max}}{P}$$
where $I_{\mathrm{mod},\max}$ is the maximum output current to the matching load, *P* represents the total power consumption. Despite fabrication process limitations, our design achieves competitive performance metrics and introduces a versatile output driver architecture suitable for multi-scenario optical communication systems.

Based on the performance summary in Table 4, the proposed PBT-based driver exhibits competitive power efficiency and compact chip area compared to prior PBT implementations. Fabricated in a 180 nm CMOS process and operating at 10 Gbps, it achieves a modulation current of up to 40 ${\mathrm{mA}}_{\mathrm{pp}}$ with a power consumption as low as 260 mW in differential 50-Ω mode—significantly lower than the 1116 mW reported in [17]. Furthermore, it supports three impedance modes compared to the single-mode operation of conventional PBT drivers, improving flexibility without incurring area overhead.

The proposed ABT driver also demonstrates notable advantages over existing ABT designs. It delivers a higher maximum modulation current of 80 ${\mathrm{mA}}_{\mathrm{pp}}$, outperforming the 60 ${\mathrm{mA}}_{\mathrm{pp}}$ in [18] and 5 ${\mathrm{mA}}_{\mathrm{pp}}$ in [19], while maintaining competitive power consumption between 340 mW and 460 mW. With the smallest reported area among cited ABT designs and a peak FoM exceeding 0.174 in differential 50-Ω mode, it achieves an effective balance between performance, configurability, and integration cost.

Although implemented in a mature 180 nm CMOS process, the proposed driver architectures are highly portable to advanced technology nodes, where higher transit frequencies would further unlock the bandwidth potential of both PBT and ABT topologies. This scalability, combined with multi-mode reconfigurability and broad compatibility with diverse optical devices, enables flexible deployment in AI-driven optical interconnects that demand adaptive, high-throughput, and energy-efficient transmitter solutions.

## 5. Conclusions

In conclusion, this work presents a versatile output driver IC design approach that respectively adopts passive (PBT) and active back termination (ABT) techniques. Fabricated in a standard 180-nm CMOS process, the two driver implementations respectively support three manually selectable impedance modes–open-drain, differential 50-Ω, and differential 100-Ω back termination–enabling broad compatibility with various optical modulators. Measurements show the effectiveness of both the passive and active back termination topologies, each providing specific advantages in terms of bandwidth, power efficiency, and output swing. These features enhance the driver’s adaptability, making it well-suited for diverse optical communication applications.

## Figures and Tables

**Figure 1 micromachines-16-01262-f001:**
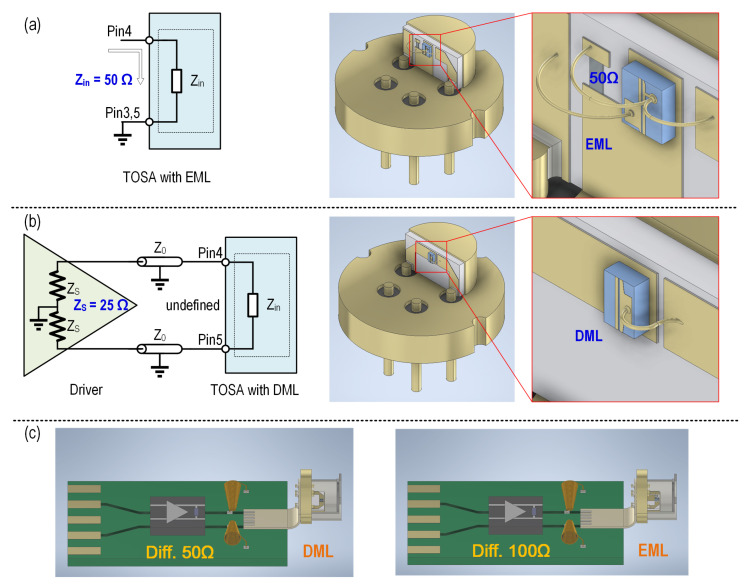
MSA definition of TOSA impedance control demands and TO package of (**a**) EML and (**b**) DML with (**c**) their application structure diagrams.

**Figure 2 micromachines-16-01262-f002:**
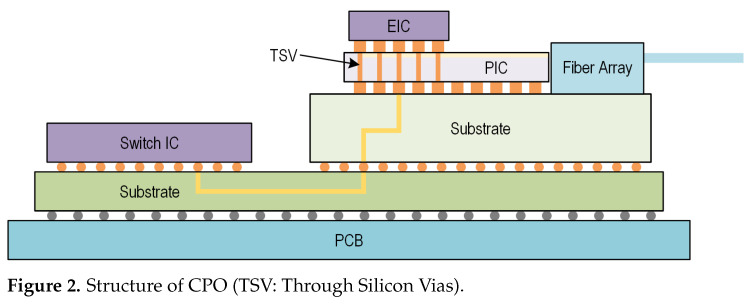
Structure of CPO (TSV: Through Silicon Vias).

**Figure 3 micromachines-16-01262-f003:**
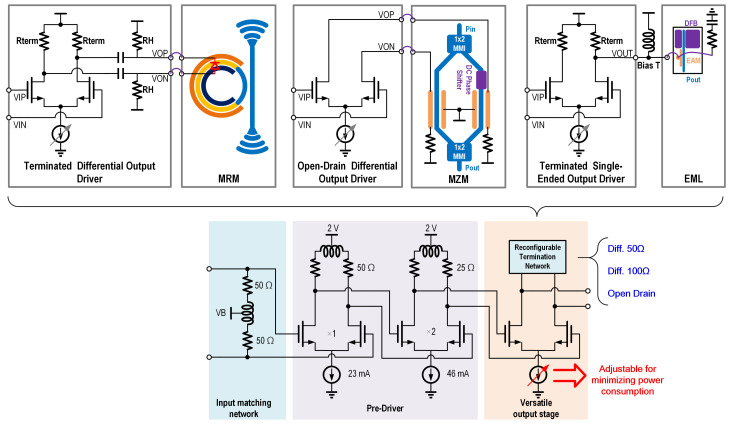
Schematic of the proposed versatile output driver.

**Figure 4 micromachines-16-01262-f004:**
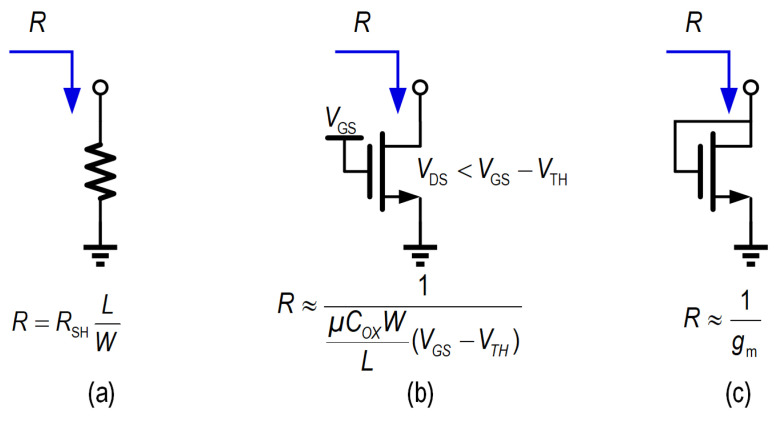
Components for passive back termination: (**a**) Resistor; (**b**) Linear-region MOSFET; (**c**) Diode-connected MOSFET.

**Figure 5 micromachines-16-01262-f005:**
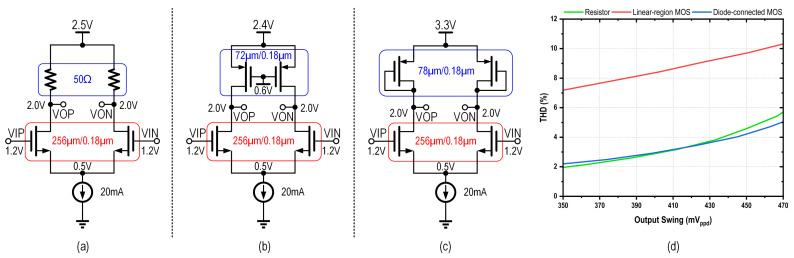
Schematic of the 6-dB-gain amplifier with different types of loads: (**a**) Diffused resistor; (**b**) Linear-region MOSFET; (**c**) Diode-connected MOSFET; (**d**) their simulated THD versus output swing.

**Figure 6 micromachines-16-01262-f006:**
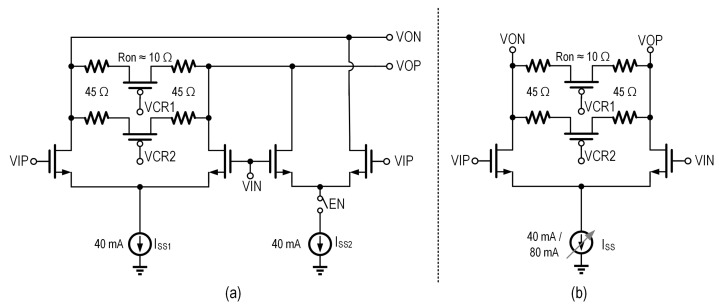
Schematics of the reconfigurable output stage with PBT, employing N-channel MOSFET (NMOS) with (**a**) constant (**b**) variable current density.

**Figure 7 micromachines-16-01262-f007:**
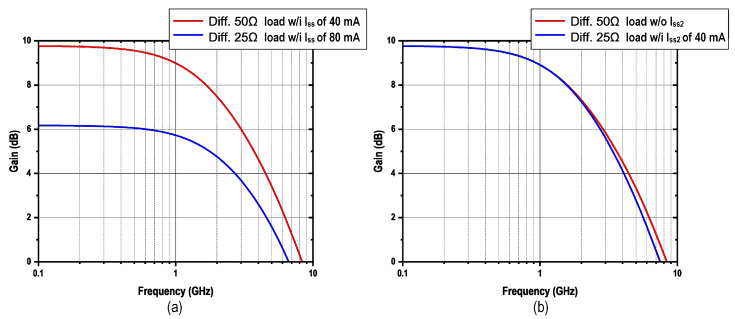
Simulated AC transfer function of the reconfigurable PBT output stage with (**a**) variable current density and (**b**) constant current density.

**Figure 8 micromachines-16-01262-f008:**
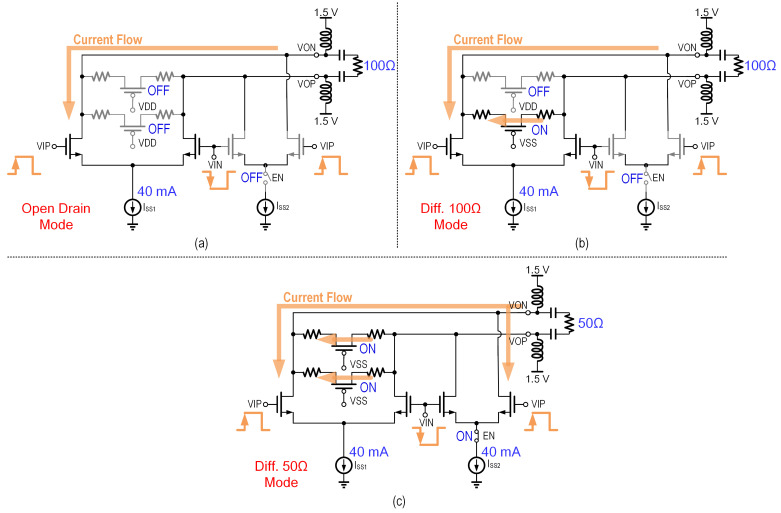
Switch status of different modes of the versatile output stage with PBT: (**a**) open-drain mode; (**b**) differential 100-Ω mode; (**c**) differential 50-Ω mode.

**Figure 9 micromachines-16-01262-f009:**
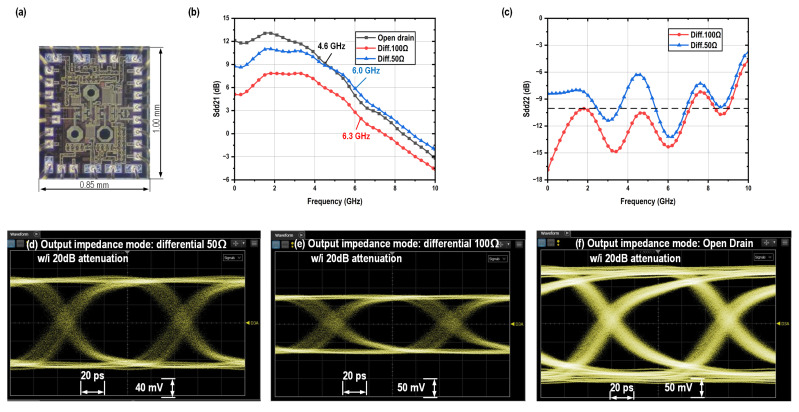
(**a**) Micro photograph of the PBT based VOD and its measurement results: (**b**,**c**) S parameters and (**d**–**f**) eye diagrams.

**Figure 10 micromachines-16-01262-f010:**
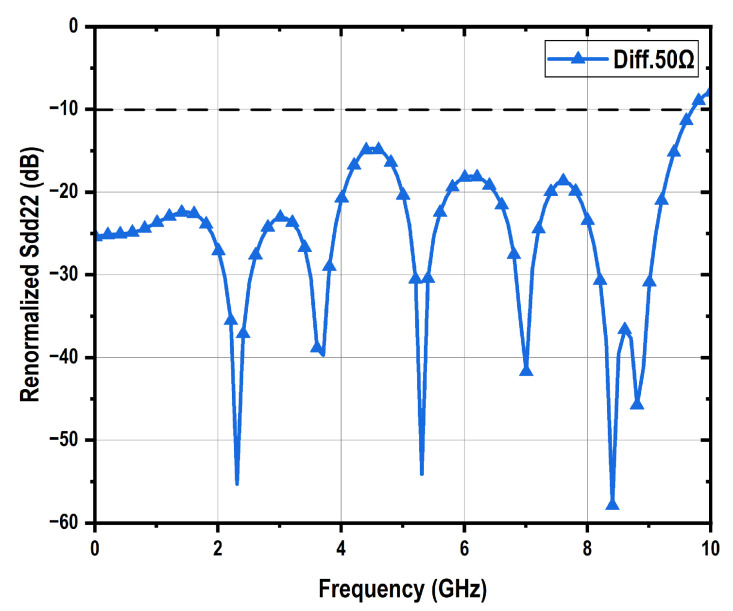
Renormalized Sdd22 of PBT based VOD under differential 50-Ω mode.

**Figure 11 micromachines-16-01262-f011:**
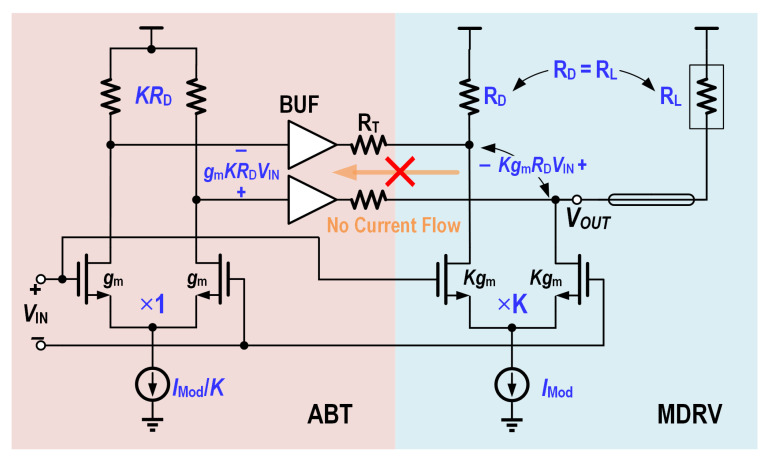
Schematic of the driver output stage with ABT.

**Figure 12 micromachines-16-01262-f012:**
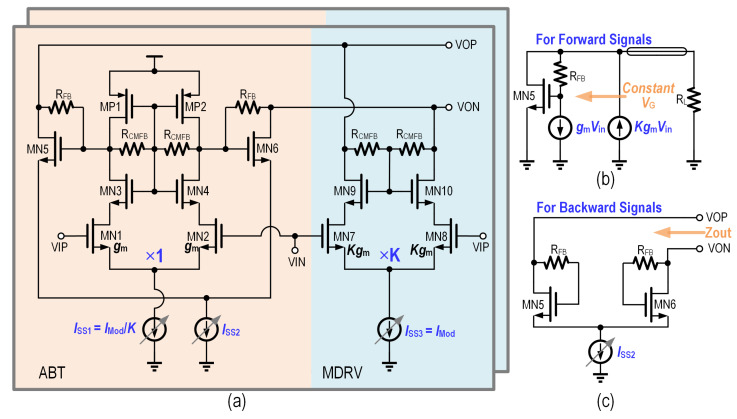
(**a**) Schematics of the reconfigurable output stage with ABT and its equivalent half circuit for (**b**) forward and (**c**) backward signals.

**Figure 13 micromachines-16-01262-f013:**
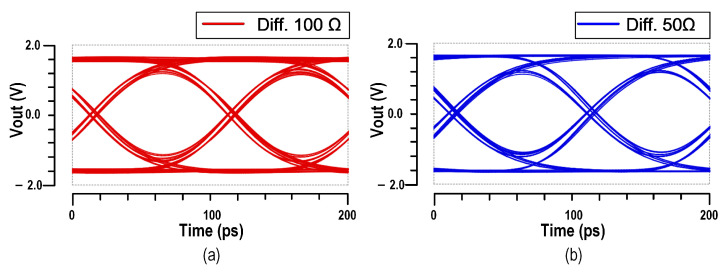
Simulated results of eye diagrams of the ABT based VOD: (**a**) differential 100-Ω mode; (**b**) differential 50-Ω mode.

**Figure 14 micromachines-16-01262-f014:**
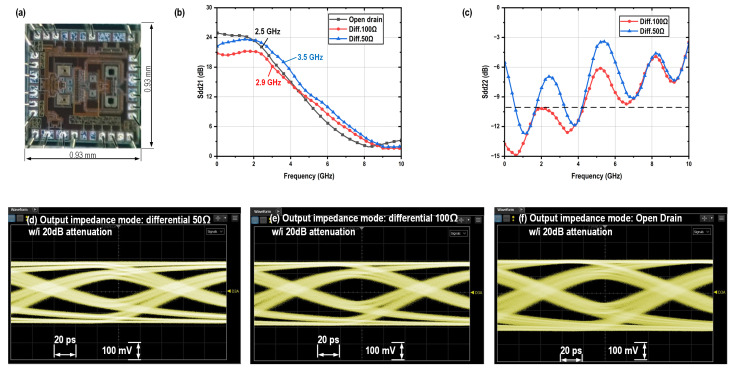
(**a**) Micro photograph of the ABT based VOD and its measurement results: (**b**,**c**) S parameters and (**d**–**f**) eye diagrams.

**Figure 15 micromachines-16-01262-f015:**
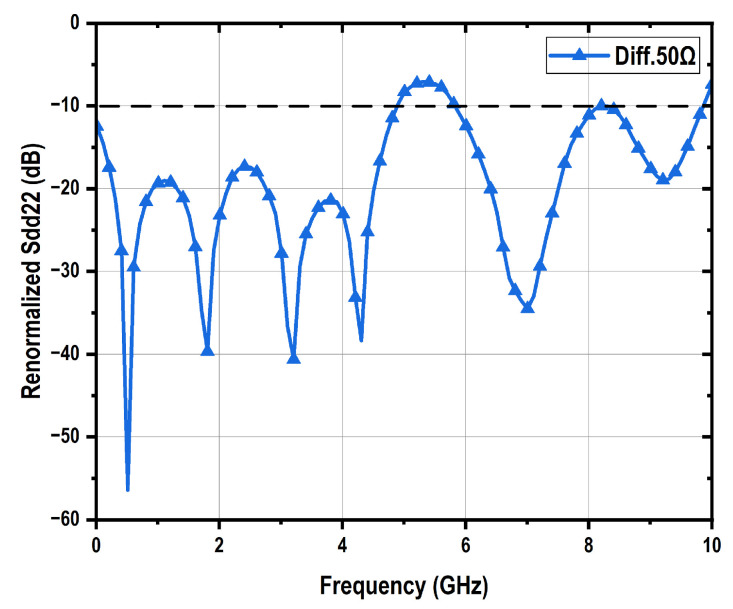
Renormalized Sdd22 of ABT based VOD under differential 50-Ω mode.

**Table 1 micromachines-16-01262-t001:** Comparison of Direct and External Modulation Devices.

Device	VCSEL	DFB	EML	MZM
Modulation Type	Direct	Direct	External	External
Impedance Requirement *	-	25 Ω	50 Ω	50 Ω
Drive Swing	<10 mA	∼40 mA	1–2 V	3–5 V
Power Consumption	Very Low	Low	Moderate	High
Chirp	Moderate	High	Low	Very Low
Launch Power	Low	Medium	Medium	High
Transmission Range	<2 km	∼10 km	∼40 km	>100 km

* For single-ended driving TOSA packaging.

**Table 2 micromachines-16-01262-t002:** Performance Comparison of Resistor Types for Passive Back Termination.

Resistor Type	Diffused	Polysilicon	Metal
Sheet Resistance (Ω/sq)	∼20 ^†^	∼600 ^‡^	<1
Temperature Stability	Poor	Good	Poor
Process Variation	Large	Moderate	Small
Accuracy	Low	Moderate	High
Current Density Tolerance (mA/μm)	∼5 ^†^	∼0.4 ^‡^	∼1
Area Efficiency *	Good	Poor	Poor

^†^ With silicide; ^‡^ Without silicide; * For 50-Ω resistors with current density tolerance of 10 mA.

**Table 3 micromachines-16-01262-t003:** Performance and Application Comparison: ABT vs. PBT.

	PBT	ABT
Power Efficiency	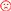 Low (Theoretical max ≤ 50%)	☺ High (>77% total output stage efficiency in this work)
Bandwidth	☺ High (Simple structure, lower parasitics)	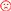 Medium (Limited by active circuitry and efficiency-bandwidth trade-off)
Output Swing	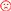 Varies with operating mode	☺ Stable across different impedance modes
Design Complexity	☺ Low (Simple structure, easy implementation)	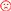 High (Complex design, requires stability consideration)
Typical Application	Ultra high speed, long-reach systems	High-density, power-sensitive systems
Key Advantage	Bandwidth, Design Simplicity	Power Efficiency, Stable Output Swing

**Table 4 micromachines-16-01262-t004:** Comparison with the state of the art.

	[15]	[16]	[17]	[18]	[19]	This Work					
Process	180 nm CMOS	28 nm CMOS	180 nm SiGe	65 nm CMOS	40 nm CMOS	180 nm CMOS					
Modulation Format	NRZ	NRZ	NRZ	NRZ	PAM4	NRZ					
Data Rate (Gbps)	12.5	12.5	10	32	32	10					
Termination Type	PBT	PBT	PBT	ABT	ABT	PBT			ABT		
Impedance Mode	1	1	1	1	1	3					
						50 Ω	100 Ω	OD	50 Ω	100 Ω	OD
$I_{\mathrm{mod},\max}\left({\mathrm{mA}}_{\mathrm{pp}}\right)$ ^#^	12	N.A.	N.A.	60	5	40	20	40	80		
Area (${\mathrm{mm}}^{2}$)	0.21	3.76 *	2.85	1.20	0.87	0.85			0.87		
Power (mW) ^‡^	142	335	1116 ^†^	550	146.8	260	200	200	460	400	340
FoM (A/W)	0.085	N.A.	N.A.	0.109	0.034	0.154	0.100	0.200	0.174	0.200	0.235

* Quad area. ^†^ Laser diode on. ^#^ With a load matching the driver’s output impedance. ^‡^ With maximum $I_{\mathrm{mod}}$. NRZ: Non-Return-to-Zero.

## Data Availability

Data are contained within the article.

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
