# Peer review of "Synergistic Driver-Laser/Modulator Co-Design with Versatile Output Stage: A Unified Optical Transmitter EIC Design Approach"

_micromachines, 2025, doi:10.3390/mi16111262_

Round 1

Reviewer 1 Report

Comments and Suggestions for Authors

This work proposes a unified optical transmitter EIC design to address the power challenges of AI data centers. This article is well-written and well-organized; I recommend it for publication after minor revision. Below, I explain my concerns and suggestions; the authors are invited to address my questions and doubts in their responses.

Author Response

Dear Reviewer,
Thank you very much for your thorough review and valuable comments and 
suggestions on our manuscript. Your insightful feedback has been instrumental 
in improving the quality of our paper. We have carefully addressed each of your 
points as detailed in the attachment.

Reviewer 2 Report

Comments and Suggestions for Authors

The manuscript entitled “Synergistic driver-laser/modulator co-design with versatile output stage: A unified optical transmitter EIC design approach” reports on the unified optical transmitter EIC design approach featuring synergistic driver-laser/modulator co-design and a versatile output driver (VOD). The paper is generally well organized and technically relevant; however, several issues in presentation and clarity should be addressed before publication. Here are the comments on this manuscript.  

  1. Some abbreviations are not defined in the manuscript. For instance, “EIC” (Line 6) and “DML” (Line 94) require definitions. In addition, “mosfet” should be corrected to “MOSFET.” The term “peak-fT” (Line 187) should be written consistently. The authors should also avoid redefining abbreviations multiple times; each abbreviation should be introduced only once at its first appearance.
  2. There is repetition between Lines 79 and 85. Figure citations also lack clarity; for example, the reference in Line 93 should read “Fig. 1a” instead of “Fig. 1.”
  3. The reference for Table 1 is not cited
  4. The unit in Table 2 is not visible for “sheet Resistance”
  5. The “simulated total harmonic distortion (THD) versus output swing” shown in 5b requires a more detailed discussion. The authors should elaborate on the observed trends and their implications for the proposed design.
  6. The author said in line 160. “Diffused resistors maintain the best linearity–efficiency trade-off among passive termination options.” However, the diode-connected MOS configuration also demonstrates comparable performance. This point should be addressed and discussed for completeness.

Author Response

Dear Reviewer,

Thank you very much for your thoughtful review and positive feedback on our manuscript entitled “Synergistic driver-laser/modulator co-design with versatile output stage: A unified optical transmitter EIC design approach”. We sincerely appreciate your valuable comments and suggestions, which have helped us significantly improve the clarity and technical depth of the paper. We have carefully addressed all the points you raised, as detailed below.

Point 1: Some abbreviations are not defined in the manuscript. For instance, “EIC” (Line 6) and “DML” (Line 94) require definitions. In addition, “mosfet” should be corrected to “MOSFET.” The term “peak-fT” (Line 187) should be written consistently. The authors should also avoid redefining abbreviations multiple times; each abbreviation should be introduced only once at its first appearance.

Response: Thank you for pointing out these omissions and inconsistencies. We have made the following corrections:

  • “EIC” is now defined as “Electronic Integrated Circuit” at its first occurrence in Line 6.
  • “DML” is now defined as “Directly Modulated Laser” when it first appears in Line 94.
  • All instances of “mosfet” have been corrected to “MOSFET”.
  • The term “peak-fT” has been standardized as “at the peak transit frequency” in the manuscript.
  • We have carefully reviewed the entire document to ensure that each acronym is defined only once at its first use.

Point 2: There is repetition between Lines 79 and 85. Figure citations also lack clarity; for example, the reference in Line 93 should read “Fig. 1a” instead of “Fig. 1.”

Response: We have revised the repetitive content between Lines 79 and 85 to improve the flow and eliminate redundancy. Additionally, all figure citations have been checked and corrected for accuracy. The reference in Line 93 now correctly points to “Fig. 1a”.

Point 3: The reference for Table 1 is not cited.

Response: Thank you for bringing this to our attention. We have now added its reference.

Point 4: The unit in Table 2 is not visible for “sheet Resistance”.

Response: The unit for “Sheet Resistance” in Table 2 has been clearly stated as “Ω/sq”.

Point 5: The “simulated total harmonic distortion (THD) versus output swing” shown in 5b requires a more detailed discussion. The authors should elaborate on the observed trends and their implications for the proposed design.

Response: We have expanded the discussion related to Fig. 5b. The revised text now explains the trend of increasing THD with output swing, relates it to the nonlinear behavior of the gain stage with different types of loads, and discusses the implications for maintaining signal integrity while optimizing power efficiency in the proposed design.

Point 6: The author said in line 160: “Diffused resistors maintain the best linearity–efficiency trade-off among passive termination options.” However, the diode-connected MOS configuration also demonstrates comparable performance. This point should be addressed and discussed for completeness.

Response: We agree with that the diode-connected MOS configuration demonstrates comparable THD. However, it shows a trade-off in power efficiency. The diode-connected MOS configuration consumes higher power with higher power supply requirement shown in Fig. 5c.

Once again, we extend our sincere gratitude for your constructive and insightful comments, which have greatly enhanced the quality of our manuscript. We hope that the revisions meet with your approval.

Sincerely,
The Authors